# Exploring multistability in prismatic metamaterials through local actuation

Agustin Iniguez-Rabago [1], Yun Li [1] & Johannes T.B. Overvelde [1]*

Metamaterials are artificial materials that derive their unusual properties from their periodic architecture. Some metamaterials can deform their internal structure to switch between different properties. However, the precise control of these deformations remains a challenge, as these structures often exhibit non-linear mechanical behavior. We introduce a computational and experimental strategy to explore the folding behavior of a range of 3D prismatic building blocks that exhibit controllable multifunctionality. By applying local actuation patterns, we are able to explore and visualize their complex mechanical behavior. We find a vast and discrete set of mechanically stable configurations, that arise from local minima in their elastic energy. Additionally these building blocks can be assembled into metamaterials that exhibit similar behavior. The mechanical principles on which the multistable behavior is based are scale-independent, making our designs candidates for e.g., reconfigurable acoustic wave guides, microelectronic mechanical systems and energy storage systems.

---

[1] AMOLF, Science Park 104, 1098 XG Amsterdam, Netherlands. *email: overvelde@amolf.nl

In the search for materials with exotic properties, researchers have recently started to explore the design of their mesoscopic architecture[1]. These so-called metamaterials have properties that arise not only from their chemical composition, but rather from the interplay between stimuli and the material's periodic structure. Examples include auxetic behavior[2,3], mechanical cloaking[4] and non-reciprocal response[5]. A challenging problem has been to design multifunctional materials, i.e., materials that can vary their properties. So far, this has been achieved by e.g., taking inspiration from origami to create internal structures that can be reconfigured along a few degrees of freedom[6–8]. Finding the structure of such reconfigurable materials is not trivial since the number of degrees of freedom for a general origami design grows exponentially[9], and typically a general design approach[10] is needed to satisfy required conditions[11]. Once created, these materials exhibit highly anisotropic behavior, enabling the change of their properties by applying locally a range of stimuli including air pressure[12,13], pre-stresses[8], and swelling[14]. However, the deformed state of these materials becomes dependent on these stimuli, and once they are removed the material will relax to the initial configuration.

A way to overcome this dependency is to introduce multi-stability[15–20]. This can be achieved by having two or more stable states that differ in configuration and are separated by significant energy barriers. Multistability has already been used to create auxetic[2,21] and energy trapping metamaterials[22–24] as well as deployable[20,25], morphing[26] or crawling[27] structures, however, most of these materials are assembled from 2D building blocks that can switch between only two stable states. A natural question to ask is whether 3D building blocks with more than two stable configurations exist and if they can be used to form multi-functional metamaterials.

As such, here we study a class of prismatic multistable 3D building blocks, that are based on polyhedra templates. These building blocks have previously been studied from an infinitesimal deformation and rigid origami framework[6]; however here we assume that these structures can undergo large rotations and deformations of the faces, making the energy of these structures highly non-linear and thus significantly more difficult to explore. To do so, we introduce a numerical method to search for energy minima that correspond to the stable states of the prismatic structures. While a complete description of all possible deformations and stable states is not possible due to the large number of degrees of freedom arising from the elastic description, our method was designed to closely mimic possible experimental implementations of locally actuated metamaterials previously studied for only one prismatic structure[12]. As a result, we are able to shine light on the highly multistable behavior that most of these building blocks exhibit. We start by introducing the design of the 3D building blocks and a numerical model to simulate their behavior. We next validate our numerical approach with centimeter-scale prototypes. In order to gain insight in the problem, we then visualize the non-linear energy landscape of multiple prismatic building blocks by applying local actuation to two hinges. Based on these results, we develop a method to extract all possible unique actuation patterns, allowing us to efficiently scan through the energy landscape and find additional stable configurations. Finally, we show for a few multistable building blocks that they can be tessellated to create multistable metamaterials.

## Results

**Design.** The structures investigated here are constructed based on templates of space-filling tessellations of polyhedra[28]. Each polyhedron in the tessellation is used as a basis for a thin-walled building block, that is constructed by extruding the edges of the

polyhedron in the direction normal to the corresponding face (Fig. 1). When assuming rigid origami[29] (i.e., the structure can only fold along predefined hinges), some of the building blocks cannot change shape (Fig. 1a, b), while others can be reconfigured along specific degrees of freedom (Fig. 1c). Interestingly, for all of these examples we found additional stable configurations that are spatially admissible, but that cannot be reached without temporarily deforming the rigid faces (Fig. 1). Under the assumption of rigid origami, these states correspond to minima in elastic energy that are separated by infinite high barriers, i.e., they are topologically isolated[30]. By allowing the faces to stretch or bend (i.e., elastic origami)[17], we lower the energy barrier such that moving between local minima becomes kinetically admissible. Note that under the assumption of elastic origami the structure has many degrees of freedom, however, some deformations require significantly less energy than others corresponding to the degrees of freedom obtained when assuming rigid origami. We refer to these deformations as soft modes instead of degrees of freedom. While for some simple origami patterns the energy of the system can be computed analytically[15,16,31], already a generic degree-four vertex pattern becomes nearly impossible to decipher[19]. The 3D prismatic structures considered in this study are constructed from non-flat degree-six, degree-eight, and degree-ten vertices, and therefore an efficient numerical technique is needed to explore the energy landscape and discover new stable states.

To model the thin-walled prismatic structures, as shown in Fig. 1, we define the elastic energy of the prismatic structures using linear springs similar to previous work[16,17,32] (Methods). Each hinge is modeled as a torsional spring with stiffness $k_h$, in which contact is taken into account by constraining the angle between $-\pi \leq \theta \leq \pi$. We account for in-plane stretching of the faces by applying springs with stiffness $k_s$ along the edges and the diagonals[32]. Bending of the faces is prevented using a set of constraints (Methods). As such, the relation between face deformation and hinge bending is specified by the ratio $\kappa = k_h/k_s$. Note that when simulating origami, deformation of the faces is typically modeled using bending instead of stretching[16]. However, to reduce the computational requirement, the observed deformation in our prototypes can be approximated using only in-plane stretching that is the result of the stretchability and flexibility of the

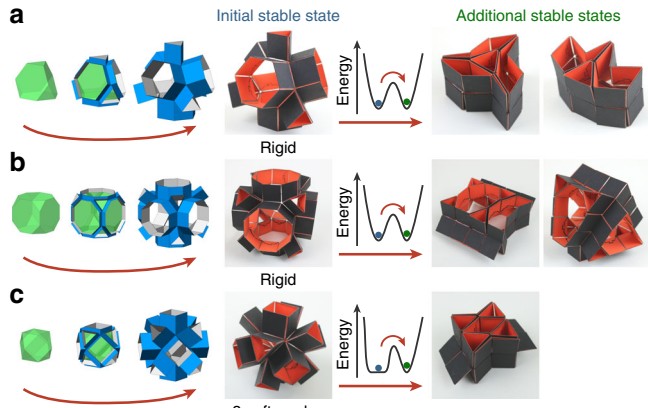

**Fig. 1 Prismatic structures and some of their stable states.** The prismatic structures can be designed by extruding the edges of a convex polyhedron perpendicular to the faces. The multistable examples shown here are based on **a** a truncated tetrahedron, **b** a truncated cube and **c** a cuboctahedron. The additional stable states can only be reached by going over a finite energy barrier resulting from deformation of the faces of the structure. The prototypes have square faces of 24 mm made from cardboard (0.4 mm thick) and connected through hinges made from double-sided tape[6].

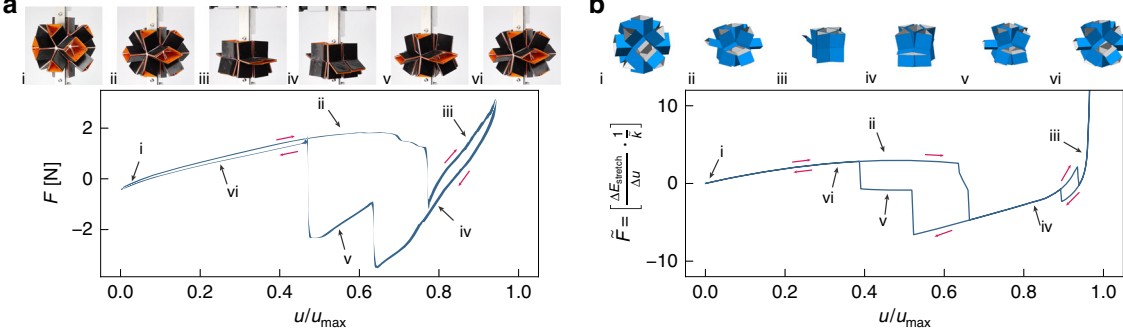

**Fig. 2 Comparison of multistable behavior between experimental and numerical compression tests. a** Average force–displacement response obtained by cyclic loading (five times) of an experimental prototype of a prismatic structure based on a cuboctahedron made from 3D printed faces (2 × 0.5 mm thick PLA) connected by flexible Mylar hinges (50 μm thick). The standard deviation is indicated by the thickness of the line. The force changes sign when an energy barrier is crossed, indicating that the structure deforms from one stable state to another. **b** Normalized force–displacement response obtained with our numerical method for the same structure. The behavior of both the experiments and the simulations are strikingly similar, both showing one instability when compressing the structure and two when returning to the initial configuration. Note that a small hysteresis loop observed in simulations at the end of the compression is not captured in experiments possibly due to a small misalignment when clamping the structure or small fabrication errors of the prototype.

hinges as observed in experiments[20]. Additionally, this simplifies the problem by reducing the number of parameters in the simulations (i.e., bending stiffness of the faces is not considered).

**Compression experiments**. We first verify the numerical model with compression experiments. We performed experiments on a prismatic structure based on a cuboctahedron, as the shape of two of the stable states (Fig. 1b) is compatible with the compression applied along a specific axis (Supplementary Movie 1). Note that this compression does not undergo any deformation along the three soft modes that this structure exhibits, therefore implying deformation of the faces. To fabricate the prototype, we built each face from two 3D-printed (Ultimaker 3) lego-like building blocks (0.5 mm thick polylactide, PLA), between which a thin lasercut sheet (50 μm thick Mylar) was manually clamped to connect the faces and form the flexible hinges. To compress the structure between two stable states we used a material test machine (Instron 5965L9510), in which we applied a cyclic displacement with a magnitude of $u_{max} = \sqrt{2}L$. Figure 2a shows the average force–displacement response of the last five compression cycles, in which the line thickness indicates the standard deviation.

The results show an initial increase in force ($F$) due to the elastic deformation of the structure. At $u/u_{max} \approx 0.6$ the response reaches a plateau, after which instabilities of the structure start to appear indicated by sudden drops in the reaction force. Importantly, the force reaches negative values after the instability at $u/u_{max} \approx 0.75$, implying that the structure has passed an energy barrier reaching a different stable state. When slowly releasing the compression, the structure follows a different path as can be seen from the hysteresis. Simulations of the same loading conditions using our numerical model are shown in Fig. 2b (Methods). While some differences exist between simulations and experiments, the similarity is striking. Both the experimental and numerical response are characterized by the same instabilities and deformation sequences (see insets in Fig. 2). Moreover, additional simulations (Supplementary Fig. 1) reveal that the observed response is mainly due to stretching of the faces and not bending of the hinges, since the stiffness ratio $\kappa$ can be increased by at least one order of magnitude (from $\kappa = 10^{-4}$ to $\kappa = 10^{-3}$) without seeing any major effect on the response. These experiments show that the simulations can qualitatively predict the behavior of the structure. Therefore, our numerical

models can be used to find stable states for the prismatic geometries.

**Applying local actuation**. To gain insight into the non-linear behavior of these structures when applying local actuation, we next visualize a 2D projection of the energy landscape that can be obtained by actuating hinge-pairs. The actuation is achieved in our simulations by applying torques to the specific hinges, forcing them towards a target angle. Figure 3a shows the energy of a prismatic structure based on a triangular prism, where we first deform the structure by actuating hinge $b$ to $\theta_b$, after which we actuate hinge $a$ to $\theta_a$. The energy is normalized by the maximum folding of the hinges and stretching of the faces (Methods). Interestingly, for $\theta_a + \theta_b < \pi$ deformation is dominated by folding of the hinges, indicating that the structure is rigidly foldable (i.e., deforms along one of the two soft modes). However, for $\theta_a + \theta_b > \pi$ the faces of the structure start to deform, leading to a dramatic increase in the elastic energy. For larger deformations the projection of the elastic energy becomes discontinuous, indicated by sudden drops, such that the structure undergoes instabilities during loading.

After deforming the structure, we release both torques and let the structure relax to a local energy minimum (Supplementary Movie 2). If the configuration after relaxation is different than the initial state, we have found a new stable state. The state diagram in Fig. 3b shows that for the structure based on a triangular prism, all initially deformed states converge to two stable states with $(\theta_a, \theta_b) = (2/3\pi, 2/3\pi)$ and $(\pi, 0)$ indicated by the star symbols in Fig. 3b. Repeating this analysis for all other hinge-pairs reveals the existence of at least three stable configurations (Supplementary Fig. 2).

Similarly, we can apply this analysis to different prismatic structures, for example one based on a truncated tetrahedron (Fig. 3c, d). This structure does not exhibit any soft modes in its initial configuration, as expected from the initial experiments (Fig. 1) and previous numerical studies[6]. Nevertheless, we find several stable states. Already when actuating one hinge-pair, the simulations reveal a highly complex energy landscape with pathways that lead to 16 stable states. Some of these are related by rotational or mirror symmetries. In order to consider these symmetries we create a clustering method based on the values of the angles of all the hinges in the stable state to find the unique stable states (Methods). Following this method, we only obtain

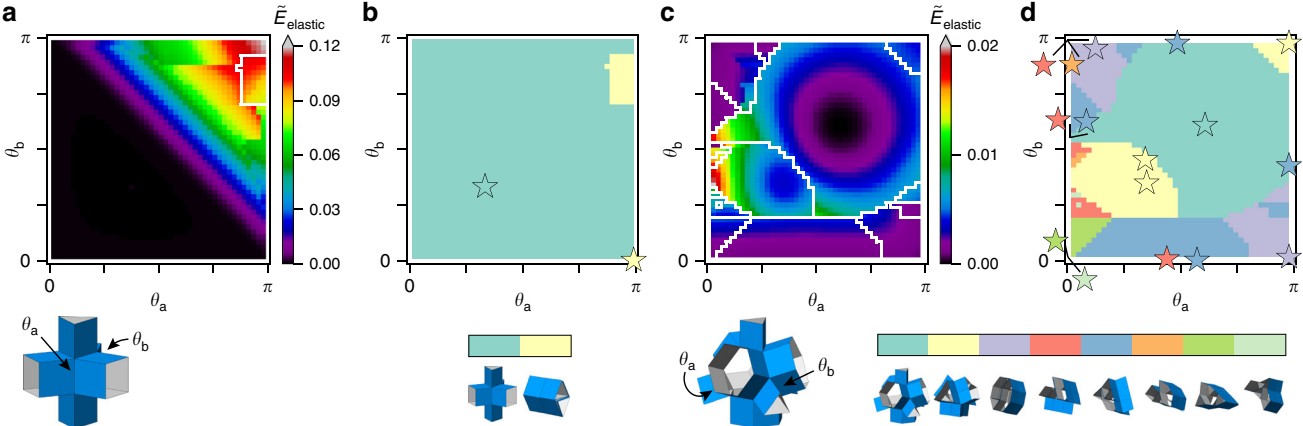

**Fig. 3 Projected energy landscapes and state diagrams for two prismatic structures.** These landscapes and states diagrams were obtained by running multiple simulations with different loads applied to two hinges. The studied prismatic structures are based on **a** a triangular prism and **c** a truncated tetrahedron. Here, we fold the structures in two steps: first we fold hinge $\theta_b$, after which we fold hinge $\theta_a$ while keeping $\theta_b$ fixed. We record the elastic energy of the final state, and repeat this process for all possible combination of angles. The results are then presented as a 2D projection of the energy landscape. The corresponding state diagrams are shown in **b** and **d**, which are obtained by relaxing the prismatic structure from the folded configurations. The star symbols indicate the values of the angles $\theta_a$ and $\theta_b$ in the final configuration.

eight unique stable states. It is interesting to note that $(\theta_a, \theta_b) = (0, \pi)$ (Fig. 3d) describes three different stable configurations. This is a clear indication that we are looking at a 2D projection of a higher dimensional energy landscape. Furthermore, by changing the order of loading (Supplementary Fig. 3), the final deformed shape of the structures varies, indicating path-dependent behavior. Finally, by analyzing all other hinge-pairs (Supplementary Fig. 3), we found a total of 12 unique stable states.

**Reducing the search space.** Selecting hinge-pairs to scan through the energy landscape is a successful method to find energy minima, however precise control of actuation applied to multiple hinges is difficult to achieve in experiments[12]. Therefore, we need a more realistic approach to search for additional states that can be achieved in experiments. To do so, we limit the actuation to either off ($\theta = 0$) or on ($\theta = \pi$). This enables us to explore a larger range of hinge combinations to which local actuation is applied, and broadens our search for other stable configurations. However, before doing so, we first describe a method to reduce the search space in order to significantly reduce the computational needs.

We start by deriving the number of hinge combinations that can be actuated. For these combinations we consider all hinges of the structure, because by definition deformation of the structure requires energy (e.g., actuation) regardless whether the structure exhibits soft modes or not. Each prismatic structure is composed of two hinge types: internal and external. The internal hinges correspond to the edges of the polyhedron that is used as a template, while the external hinges arise from the extrusion process. For example, a prismatic structure based on a tetrahedron has $n_{int} = 6$ internal hinges and $n_{ext} = 2n_{int} = 12$ external hinges. If we select one of these hinges for actuation, there are a total of $\eta_1 = n_{int} + n_{ext} = n_{tot} = 18$ different possibilities. When selecting two hinges, there are a total of $\eta_2 = n_{tot}!/(2!(n_{tot} - 2)!) = 153$ combinations that we can make. In general, we can write for a selection of $s$ actuated hinges $\eta_s = (n_{tot})!/(s!(n_{tot} - s)!)$, leading to a total number of actuation combinations equal to $\eta_{tot} = 2^{n_{tot}}$. These combinations grow exponentially with the number of edges of the prismatic structure (Supplementary Fig. 4). This makes it nearly impossible to run all different actuations for prismatic structures based on larger

polyhedra, such as a cuboctahedron ($n_{int} = 24$) that has more than $10^{21}$ combinations.

By focusing only on the internal hinges of the prismatic structures ($n_{int}$), we reduce the number of combinations $\eta_{int}$ to $2^{2n_{int}}$. Note that e.g., for a cuboctahedron template this is approximately a reduction of 14 orders of magnitude (Supplementary Fig. 4). However, we still need to reduce $\eta$ further to be able to efficiently scan the energy landscape. We therefore exploit the symmetries of the prismatic structures to remove the combinations of hinges that can be rotated or mirrored leading to exactly the same actuation patterns. To find symmetric actuation patterns, we first convert the polyhedron into a directed graph, mapping all the internal hinges to nodes on the graph. Depending on the two faces of the original polyhedron that are connected by the hinge, we give each corresponding node in the graph a specific type. For example, a hexagonal prism has the types triangle-square and square-square, while a tetrahedron only has the type triangle-triangle. We then construct the graph by connecting a directed line between two nodes if both hinges share one vertex and the internal polyhedron can rotate clockwise to the normal of the face that both hinges share. Next, we determine the minimum distance matrix between nodes[33], in which we keep track of the node type encountered when traveling along the shortest path. We extract all principal sub-matrices from the distance matrix and compare their eigenvalues and vectors to identify and remove symmetric loading cases (Methods). Using this method we can further reduce the number of hinge combinations ($\eta_{sym}$) by approximately two orders of magnitude (Supplementary Fig. 4).

**Applying the actuation patterns to find stable states.** We next use these unique hinge combinations to apply discrete (i.e., on/off) actuation to the prismatic structures in order to find their stable states. As before, for each load case we first apply a torque to the corresponding hinges, after which we release the torque and let the structure relax to equilibrium. We then follow a clustering method to find the unique stable states (Methods). Additionally, we characterize the stability of these unique stable states by stepwise increasing the stiffness of the hinges by changing $\kappa$ in our numerical model, pushing the structure back to its original undeformed state. We record the last value ($\kappa_{max}$) for which the prismatic structure remains in the stable configuration.

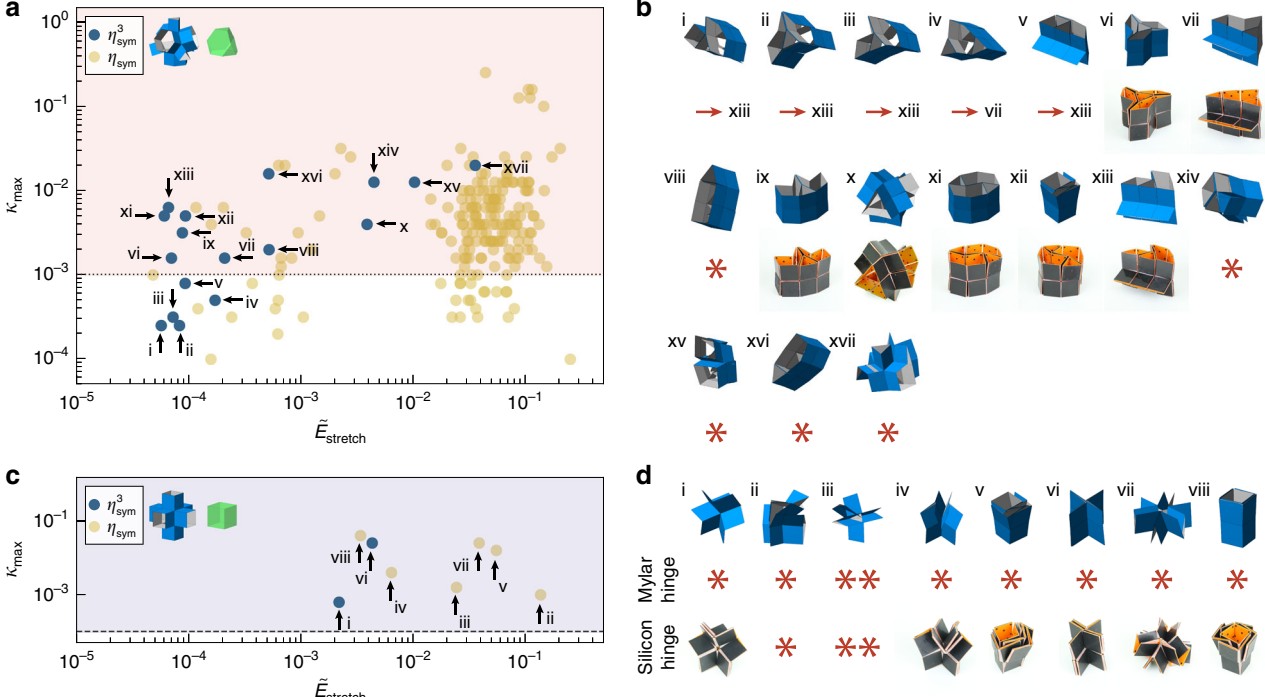

**Fig. 4 Comparison of stable states found with our numerical algorithm and experimental verification.** The normalized stretch energy ($\tilde{E}_{stretch}$) and maximum stiffness ratio ($\kappa_{max}$) were obtained with simulations for the unique stable states of a prismatic structure based on **a** a truncated tetrahedron and **c** a cube. Blue dots indicate stable states found by actuation of up to 3 hinges simultaneously, while the yellow dots indicate all other stable states found by actuating more than 3 hinges. The pink and purple colors indicate regions with stable states that were achieved in experiments. **b** A selection of the stable states of the structures based on a truncated tetrahedron is compared to experiments using a prototype with 3D printed faces and Mylar hinges. **d** The stable states of the structures based on a cube cannot be obtained with prototypes made with Mylar hinges. Instead, more stretchable silicon hinges are used to obtain the stable states. The arrows in **b** and **d** indicate that the state is not stable in experiments, but relaxes to a different state. A single asterisk (*) indicates that the stable state could not be achieved in experiments due to a limited maximum stretch, while a double asterisk (**) indicates that the state cannot be achieved due to non-adjacent face crossing.

In Fig. 4a we show the results for the prismatic geometry based on a truncated tetrahedron, in which we plot the maximum value of the hinge stiffness, $\kappa_{max}$, against the normalized stretch energy, $\tilde{E}_{stretch}$, for all the stable states that we found. While we previously found 12 stable states for this structure when actuating only hinge-pairs (Fig. 3c, d), by running all the unique hinge combinations ($\eta_{sym}$) we find a total of 213 stable states (where we already removed the duplicate and symmetric stable states). To verify these results, we performed experiments on a prototype made with the same fabrication method discussed previously (Fig. 4b). We tried to obtain the 17 states that can be reached by actuating up to 3 hinges simultaneously ($\eta_{sym}^3$). While seven states can be found directly, we observe two important differences between simulations and experiments. First, the stable states (i–v) characterized by $\kappa_{max} < 10^{-3}$ cannot maintain their stable configuration after releasing the actuation, and instead relax to states vii and xi (Supplementary Movie 3). We deduce from this observation that the value of $\kappa$ in our experiments is equal to $\kappa \approx 10^{-3}$ (dotted line in Fig. 4a). Second, the stable states viii and xiv-xvii cannot be reached in experiments due to a limitation of the maximum stretch that the hinges of the prototypes can undergo. This difference is expected, as these constraints have not been taken into account in the simulations to maximize the search space.

To highlight the influence of the maximum stretch that we observe in experiments, we also performed experiments on a prismatic geometry based on a cube. While we find a total of eight stable states in our simulations (Fig. 4c), we were not able to reach any of these configurations with the current fabrication method.

However, replacing the Mylar hinges with stretchable elastomeric hinges (0.5 mm silicon rubber) enabled us to overcome higher stretch energy barriers, such that we were able to find six of the stable states in experiments. Note that we were still not able to reach two of the stable states. The stretch to obtain state ii is still larger than the capabilities of our prototype, while state iii has non-adjacent faces crossing that are not accounted for in our numerical model. Furthermore, we find that the elastomeric hinges result in a lower $\kappa$ in the prototype, such that we were able to achieve stable states with $\kappa_{max} < 10^{-3}$.

Finally, we applied the same analysis to the other 16 prismatic structures based on regular polyhedra that either have up to $n_{int} = 30$, or that can be used to construct uniform space-filling tessellations[6]. Note that for polyhedra with $n_{int} > 18$ we were still not able to run all possible unique hinge combinations, and have limited ourselves to combinations of up to three hinges, i.e., $\eta_{sym}^3$ (Supplementary Fig. 4). The number of stable states for the studied structures ranges from 2 to 418, as reported in Supplementary Fig. 5. Furthermore, we show some of the possible stable states that we can get from two different structures based on a truncated cube and a rhombicuboctahedron in Supplementary Fig. 6.

**Multistable metamaterials.** We have shown that by local actuation we are able to effectively explore the non-linear energy landscape of prismatic structures and find additional minima. We next show that our method can also be applied to find stable states in prismatic metamaterials assembled from these building blocks, by using periodic boundary conditions (Methods). Note

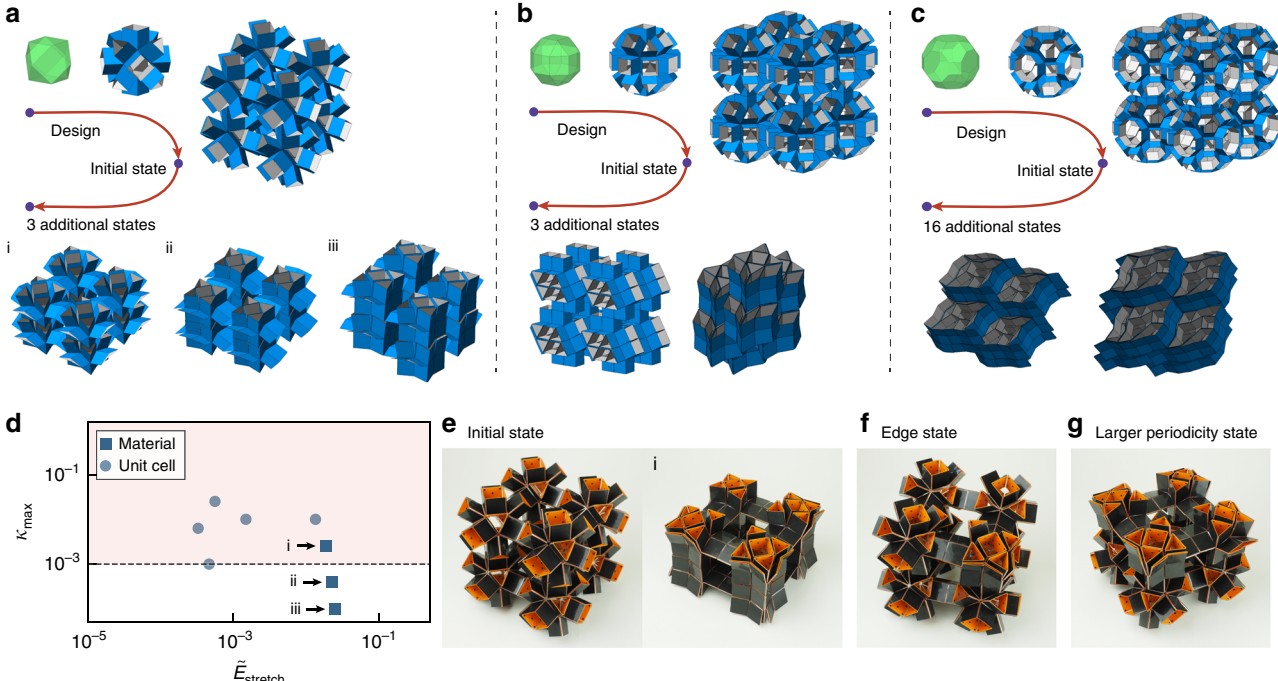

**Fig. 5 Simulations and experiments of multistable metamaterials.** The simulated metamaterials are based on cubic tesselations of prismatic structures based on **a** a cuboctahedron, **b** a rhombicuboctahedron and **c** a truncated cuboctahedron. **d** Normalized stretch energy ($\tilde{E}_{stretch}$) and maximum stiffness ratio ($\kappa_{max}$) for the unit cell and metamaterial based on a cuboctahedron. **e** One of the stable states of the prismatic structure based on a cuboctahedron found in simulations can also be achieved in a prototype of the metamaterial made with 50 μm thick Mylar hinges. Additionally, we find **f** edge and corner states and **g** stable states with periodicity larger than a single unit cell not covered by our simulations.

that periodic boundary conditions introduce additional constraints that considerably limit the stable states that can be achieved, and has a huge effect on the non-linear response of the material. In fact, many of the asymmetric stable states that we found for the unit cells can no longer be achieved in the materials. Therefore, the challenge here is to design materials that are capable of being mechanically stable in additional states. To perform a first exploration, we focus on cubic tessellations that can be constructed from a single prismatic structure. We consider 11 of the polyhedra that were previously found to be multistable, which results in a total of 15 materials (Supplementary Fig. 7). In order to search for stable states and to limit the number of actuation patterns that have to be applied, we apply the same actuation patterns to the material that previously resulted in stable configuration for the prismatic structure used as a unit cell.

We find that most (11) of the materials show additional stable states (Supplementary Fig. 7). In Fig. 5a–c and Supplementary Movie 4 we show three of these multistable materials and some of their stable states with unit cells based on a cuboctahedron, a rhombicuboctahedron and a truncated cuboctahedron. Note that the energy of the metamaterial's stable states do not have to be equal to the prismatic building blocks. This can be seen for example from the stretch energy of the stable state of the unit cell of a cuboctahedron in Fig. 5d, which have shifted for the periodic arrangements.

To validate these findings, we fabricated a material assembly based on a cuboctahedron, that contains 2 × 2 × 2 building blocks. As expected, we are only able to achieve one of the stable states (Fig. 5e), since states ii and iii lie below the $\kappa_{max} \approx 10^{-3}$ threshold as previously predicted. To verify this, we also fabricated a second sample with thicker Mylar sheets of 125 μm, and we found that this metamaterial does not exhibit any stable states (Supplementary Movie 5).

While here we have limited our analysis to periodic boundary conditions applied to the unit cells, similar as in rigid origami[6], stable states can appear on the edges and corners of the materials. This is clearly demonstrated in Fig. 5f showing an edge state. We are also able to observe additional stable states with wavelength longer than the unit cells (Fig. 5g). In order to capture these in our model, larger unit cells would have to be considered. Therefore, additional studies have to be performed to continue exploration of the rich energy landscape, including many other tessellations that can be constructed.

**Discussion**

In summary, we have introduced a computational strategy to visualize and efficiently explore the complex energy landscape of 3D prismatic structures. We revealed the vast multistability of these structures, and despite that our numerical approach only explores part of the configuration space, by basing our method on local actuation we were able to find stable states that can be achieved experimentally. Additionally, by tessellating these prismatic structures, we find multistable metamaterials that can reconfigure their architecture and therefore tune their properties. Importantly, these materials do not require energy to maintain their stable configurations, and will be robust to external variations as significant energy barriers have to be overcome to alter their structure. Moreover, by varying the relation between hinge and face stiffness, the stability of states can be tuned[20,22,34]. We believe that our local actuation strategy can also be applied to other origami-inspired metamaterials.

While we have only validated our experiments at the centimeter-scale using relatively simple fabrication techniques, the mechanical behavior of our systems is theoretically scale-independent. However fabricating such intricate structures at the micro/nano scale is not trivial. More advanced 3D-fabrication

techniques such as two-photon lithography[35] or stereo-lithography[36] could lead this innovation. Additionally, scaling of these structures will introduce environmental influences such as capillary effects. It is not known how such forces influence the (multistable) behavior of our structures, and additional research is required to explore this direction. Moreover, instead of manually deforming the structure, they can be made responsive by applying local actuation to the hinges. For our prototypes, this can be achieved by applying pneumatic pouches to some of the hinges[12]. As an example, already by discrete actuation of two pouches applied to the prismatic structure based on a truncated tetrahedron, we were able to achieve four of the stable states (Supplementary Movie 6). Similar strategies could be applied to actuate our designs at smaller length scales, using e.g., localized swelling of hydrogels for actuation. However, individually addressable local actuation patterns become difficult to apply at small scales and therefore a particular interesting future direction is the application of global stimuli to trigger the multistable behavior, e.g., heat or pH variations. Global stimuli can potentially lead to different folding behavior for a single structure by incorporating multiple materials into the design that each respond differently to changes in their environment or to different loading rates. Therefore, we believe that these prismatic multistable materials could lead to the next generation of multifunctional metamaterials that can be applied as e.g., reconfigurable acoustic wave guides[37], microelectronic mechanical systems[38] and energy storage devices[39].

## Methods

**Minimization of energy.** To simulate the folding and deformation behavior of our structures, we implemented a numerical method that minimizes the elastic energy using gradient information. First, we create the prismatic structures by extruding the faces of an internal convex polyhedron[6]. The resulting structure has $F$ rectangular faces, each surrounded by four edges ($S_E$) and divided by two diagonals ($S_D$) defined by the four vertices ($V$) of each face. Edges that connect two faces are defined as hinges ($H$). We describe the complete shape of the prismatic structure by the coordinates of its vertices $\mathbf{x} = [x_{1,1}, x_{2,1}, x_{3,1}, x_{1,2}, \dots, x_{3,V}]$. In this section, we first derive the elastic energy associated with face stretching and hinge folding, and the work applied to deform the structure. Second, we derive the required face bending, stretch and angle constraints, and the periodic boundary conditions. Third, we discuss how we normalize the energy to allow us to compare between different prismatic structures, and finally we describe the implementation of our algorithm in Matlab.

First, two types of elastic energies are assigned to the structure: the hinge folding energy $E_{\text{hinge}}$ modeled as torsional springs placed at the hinges, and the face stretch energy $E_{\text{stretch}}$ modeled as linear springs placed at the edges and the diagonals of each face. We assume that the structure has zero elastic energy in its initial extruded state with coordinates $\mathbf{X}$, such that the total elastic energy of the structure is given by

$$E_{\text{elastic}} = E_{\text{hinge}} + E_{\text{stretch}}. \tag{1}$$

The gradient of $E_{\text{elastic}}$ with respect to the displacement of the vertices $\mathbf{u} = \mathbf{x} - \mathbf{X}$ is then equal to

$$d\mathbf{E}_{\text{elastic}} = d\mathbf{E}_{\text{hinge}} + d\mathbf{E}_{\text{stretch}} = \frac{\partial E_{\text{hinge}}}{\partial \mathbf{u}} d\mathbf{u} + \frac{\partial E_{\text{stretch}}}{\partial \mathbf{u}} d\mathbf{u}. \tag{2}$$

We model each hinge as a linear torsional spring with an angle $\Theta$ in the initial state, $\theta$ in the deformed state and stiffness $k_{\text{h}}$. The hinge energy $E_{\text{hinge}}$ is defined as

$$E_{\text{hinge}}(\boldsymbol{\theta}) = \sum_{i=1}^{H} \frac{1}{2} k_{\text{h}} (\theta_i - \Theta_i)^2 = \frac{1}{2} k_{\text{h}} (\boldsymbol{\theta} - \boldsymbol{\Theta}) \cdot (\boldsymbol{\theta} - \boldsymbol{\Theta}), \tag{3}$$

where $\theta = [\theta_1, \theta_2, \dots, \theta_H]$ and $\boldsymbol{\Theta} = [\Theta_1, \Theta_2, \dots, \Theta_H]$. Each hinge angle can be found from the coordinates of the vertices according to

$$\theta = \tan^{-1} \left( \frac{\mathbf{a}_{\text{h}} \cdot (\mathbf{n}_{\text{a}} \times \mathbf{n}_{\text{b}})}{\mathbf{n}_{\text{a}} \cdot \mathbf{n}_{\text{b}}} \right), \tag{4}$$

in which $\mathbf{n}_{\text{a}}$ and $\mathbf{n}_{\text{b}}$ are the normal vectors of the two faces connected by the hinge, and the vector $\mathbf{a}_{\text{h}}$ lies along the hinge axis (Supplementary Fig. 8a). Note that we use the function $\tan^{-1}$ (instead of e.g., using $\theta = \cos^{-1}(\mathbf{n}_{\text{a}} \cdot \mathbf{n}_{\text{b}})$) since its domain is defined for $(-\infty, \infty)$ and the angle can be defined between $[-\pi, \pi]$ using a Four-quadrant inverse tangent. Partial derivatives of the hinge angles with respect to

vertex displacement can then be found according to

$$\frac{\partial E_{\text{hinge}}}{\partial \mathbf{u}} = \frac{\partial E_{\text{hinge}}}{\partial (\boldsymbol{\theta} - \boldsymbol{\Theta})} \frac{\partial (\boldsymbol{\theta} - \boldsymbol{\Theta})}{\partial \mathbf{u}} = k_{\text{h}} (\boldsymbol{\theta} - \boldsymbol{\Theta}) \mathbf{J}_{\text{hinge}}, \tag{5}$$

in which $\mathbf{J}_{\text{hinge}}$ is a Jacobian matrix with entries

$$\mathbf{J}_{\text{hinge}[i,3(v-1)+j]} = \frac{\partial \theta_i}{\partial x_{j,v}}, \tag{6}$$

for $i = 1, \dots, H, j = 1, 2, 3$ and $v = 1, \dots, V$. For a more detailed explanation and description of the Jacobian see ref. [6].

Stretching of each face is modeled using linear springs placed along the edges and diagonals. While Filipov et al.[32] derived specific expressions for the stiffness of each spring, given the constraints on out-of-plane bending of the face we simplify our model and assume that all springs have stiffness $k_{\text{s}}$. As a result, the ratio between hinge bending and face stretching is set by a single parameter $\kappa = k_{\text{h}}/k_{\text{s}}$. The face stretching energy $E_{\text{stretch}}$ can be found according to

$$E_{\text{stretch}} = \sum_{i=1}^{S_E + S_D} \frac{1}{2} k_{\text{s}} (l_i - L_i)^2 = \frac{1}{2} k_{\text{s}} (\mathbf{l} - \mathbf{L}) \cdot (\mathbf{l} - \mathbf{L}), \tag{7}$$

in which $L_i$ and $l_i$ correspond to the initial and deformed length of the i-th edge, respectively, $\mathbf{L} = [L_1, L_2, \dots, L_{S_E + S_D}]$ and $\mathbf{l} = [l_1, l_2, \dots, l_{S_E + S_D}]$. Furthermore, the change in length of each edge can be found from the displacement of the two corresponding vertex displacements (Supplementary Fig. 8b)

$$l - L = \sqrt{\sum_{i=1}^{3} (u_{i,\text{a}} - u_{i,\text{b}})^2}. \tag{8}$$

The partial derivative of the face stretching energy then equals

$$\frac{\partial E_{\text{stretch}}}{\partial \mathbf{u}} = \frac{\partial E_{\text{stretch}}}{\partial (\mathbf{l} - \mathbf{L})} \frac{\partial (\mathbf{l} - \mathbf{L})}{\partial \mathbf{u}} = k_{\text{s}} (\mathbf{l} - \mathbf{L}) \mathbf{J}_{\text{stretch}}, \tag{9}$$

where $\mathbf{J}_{\text{stretch}}$ is a Jacobian with entries

$$\mathbf{J}_{\text{stretch}[i,3(v-1)+j]} = \frac{\partial l_i}{\partial x_{j,v}}, \tag{10}$$

for $i = 1, \dots, S_E + S_D, j = 1, 2, 3$ and $v = 1, \dots, V$.

In order to deform the structure, we specify a target angle for $A$ hinges, and apply a penalty method to increase the energy by $E_{\text{load}}$, seen as work energy, in case the target angles are not satisfied. The total energy of the system $E$ that includes both the elastic deformation and the loads is then equal to

$$E = E_{\text{hinge}} + E_{\text{stretch}} + E_{\text{load}}, \tag{11}$$

in which

$$E_{\text{load}} = \sum_{i=1}^{A} \frac{1}{2} k_{\text{p}} (\theta_i - \hat{\Theta}_i)^2, \tag{12}$$

$k_{\text{p}}$ sets the stiffness of the applied penalty and $\hat{\Theta}_i$ is the target angle of the i-th hinge to which a load has been applied. Note that for $k_{\text{p}} \gg k_{\text{h}}$ we are practically constraining the angles to the target angles of the loaded hinges, while for smaller values of $k_{\text{p}}$ the target angles might not be reached.

Second, while with the derivation of $E_{\text{elastic}}$ we have specified the energy landscape of our prismatic structures, certain deformations are not admissible. We present five different constraints that we implemented to limit the deformation. Specifically we discuss how to: fix vertices to remove rigid body translations and rotation, prevent face bending, limit the hinge angle to implement contact, limit the edge stretching to prevent numerical convergence problems, and simulate infinite periodic tilings of the prismatic structures using periodic boundary conditions.

To prevent rigid body translations we select one vertex from the first face of the prismatic structure $\mathbf{X}_1$ and fix the displacement to $\mathbf{u}_1 = (0, 0, 0)$. Moreover, to avoid rigid body rotations we fix two other vertices on the same face $\mathbf{X}_2$ and $\mathbf{X}_3$ according to

$$\mathbf{u}_2 \cdot (\mathbf{X}_2 - \mathbf{X}_1) \times (\mathbf{X}_3 - \mathbf{X}_1) = 0, \tag{13}$$

$$\mathbf{u}_3 \cdot (\mathbf{X}_2 - \mathbf{X}_1) \times (\mathbf{X}_3 - \mathbf{X}_1) = 0, \tag{14}$$

$$\mathbf{u}_2 \cdot (\mathbf{X}_3 - \mathbf{X}_1) = 0. \tag{15}$$

To ensure that no face bending can occur, we impose that the out-of-plane displacement of each vertex on a face remains zero ($z_i = 0$ for $i = 1, 2, \dots, V_f$, in which $V_f$ is the total number of vertices of the f-th face). To determine the out-of-plane deformation we use two vectors $\mathbf{w}_1$ and $\mathbf{w}_2$ on a face to get its normal and project the remaining edge vectors ($\mathbf{w}_i$ for $i = 3, \dots, V_f - 1$) onto the face. For each face, we then have (Supplementary Fig. 8c)

$$z_i = \mathbf{w}_i \cdot (\mathbf{w}_1 \times \mathbf{w}_2) = 0, \tag{16}$$

for $i = 3, \dots, V_f - 1$.

If we write $\mathbf{z} = [z_1, z_2, \dots, z_{V_F}]$, in which $V_F = \sum^F (V_f - 3)$, the partial derivative of the face bending constraints with respect to the vertex displacement

equals

$$\mathbf{C}_{\text{bend}[i,3(\nu-1)+j]} = \frac{\partial z_i}{\partial x_{j,\nu}}, \tag{17}$$

for $i = 1, \dots, V_{\text{F}}$, $j = 1, 2, 3$ and $\nu = 1, \dots, V$.

We need to limit rotation of the hinge since each hinge connects to two faces which come into contact when $\theta = -\pi$ or $\pi$, representing a fully closed hinge. Because we use a four-quadrant inverse tangent to determine the angles of the hinges between two adjacent faces (Eq. (4)), the angle can vary between $-\pi \leq \theta \leq \pi$. However, when the two faces cross each other, the hinge angle will have a real value that lies outside the range of the inverse tangent. To avoid this problem, we ensure that the faces never cross by applying a tighter constraint to the hinge angles

$$-0.985\pi < \theta_i < 0.985\pi, \tag{18}$$

where the limits have been determined by running several simulations. Note that the gradient for this constraint is given by Eq. (6).

We also implemented two other precautionary measurements to avoid adjacent face crossing. First, we keep track of the angles of previous iteration step such that a sudden change in angle larger than $\pi$ indicates the crossing of the two faces. If this occurs, depending on the sign of the step, we add or subtract $2\pi$ to have the real value outside the range $-\pi \leq \theta \leq \pi$. Second, we are limiting the step size of the minimization function so that the difference in angles between steps cannot become greater than $\pi$ if no adjacent face crossing occurs.

While stretching is permitted in both simulations and experiments, our simulations shows convergence problems when higher stretches occur. For example, when the flat face becomes concave, the normal of the initially rectangular face becomes ill-defined. This could occur when the maximum strain of an edges of the faces become larger than 0.17. Nevertheless, we found this constraint too tight as it was prohibiting large deformations on the structure and the search for stable states. Therefore, we loosen this constraint in our simulations to a maximum strain of edges according to

$$-0.30 \leq \frac{l_i - L_i}{L_i} \leq 0.30, \tag{19}$$

where $l_i$ and $L_i$ are the deformed and original length of an edge or diagonal, respectively. For all simulations, this value did not produce ill-defined normals. The gradient for this constraint is given by the normalized Jacobian of Eq. (10), and equals

$$\mathbf{C}_{\text{stretch}[i,j]} = \frac{1}{L_i} \frac{\partial l_i}{\partial x_j}, \tag{20}$$

for $i = 1, \dots, S_E + S_D$ and $j = 1, \dots, V$.

To create metamaterials represented by infinite large tessellations of the prismatic structures, we apply periodic boundary conditions along the lattice vectors $\mathbf{A}_i$ for $i = 1, \dots, n_{\text{dim}}$. Depending on the number of lattice vectors ($n_{\text{dim}} \in [1, 2, 3]$), they can span the material in one, two or three dimensions. For the undeformed structure, two vertices are periodically located when

$$\mathbf{X}_{\text{b}} - \mathbf{X}_{\text{a}} = \sum_{i=1}^{n_{\text{dim}}} \alpha_i \mathbf{A}_i, \tag{21}$$

where $\mathbf{X}_{\text{a}}$ and $\mathbf{X}_{\text{b}}$ are the initial positions of the vertices, and $\alpha_i \in [-1, 0, 1]$ represents the possible linear combination of the lattice vectors for tiling the unit cell in space. Then, periodic boundary conditions can be applied to these periodically located vertex pairs according to

$$u_{\text{b}} - u_{\text{a}} = \sum_{i=1}^{n_{\text{dim}}} \alpha_i \mathbf{a}_i, \tag{22}$$

in which $\mathbf{a}_i$ represents the deformation of the lattice vector.

Third, to compare the results between prismatic structures based on different polyhedra, we normalize the elastic energy. The elastic energy of any structure is given by Eq. (1). Note that the constraints Eq. (18) and Eq. (19) limit the deformation of the hinge angles $\theta_i$ and the edge length $l_i$, and therefore we can define a total maximum energy when all the angles and edge lengths are at their maximum, respectively $\theta_i^{\max}$ and $l_i^{\max}$. The maximum hinge energy is then equal to

$$E_{\text{hinge}}^{\max}(\boldsymbol{\theta}) = \sum_{i=1}^{H} \frac{1}{2} k_{\text{h}} (\theta_i^{\max} - \Theta_i)^2 = \frac{1}{2} k_{\text{h}} (\boldsymbol{\theta}^{\max} - \boldsymbol{\Theta}) \cdot (\boldsymbol{\theta}^{\max} - \boldsymbol{\Theta}). \tag{23}$$

Similarly, the maximum energy as a result of in-plane face deformation equals

$$E_{\text{stretch}}^{\max} = \sum_{i=1}^{S_E + S_D} \frac{1}{2} k_{\text{s}} (l_i^{\max} - L_i)^2 = \frac{1}{2} k_{\text{s}} (\mathbf{l}^{\max} - \mathbf{L}) \cdot (\mathbf{l}^{\max} - \mathbf{L}). \tag{24}$$

We then define the total normalized elastic energy $\tilde{E}_{\text{elastic}}$ according to

$$\tilde{E}_{\text{elastic}} = \frac{E_{\text{elastic}}}{E_{\text{elastic}}^{\max}} = \frac{E_{\text{elastic}}}{E_{\text{hinge}}^{\max} + E_{\text{stretch}}^{\max}}, \tag{25}$$

and the individual components of the elastic energy are given by

$$\tilde{E}_{\text{hinge}} = \frac{E_{\text{hinge}}}{E_{\text{hinge}}^{\max}} \tag{26}$$

$$\tilde{E}_{\text{stretch}} = \frac{E_{\text{stretch}}}{E_{\text{stretch}}^{\max}}. \tag{27}$$

Last, we implement the aforementioned set of equations in MATLAB, and use the build-in non-linear constrained optimization function called 'fmincon', to minimize the elastic energy (Eq. (11)) given a set of linear (Eqs. (13), (14), (15) and (22)) and non-linear (Eqs. (16), (18) and (19)) constraints. Specifically, we choose the 'Active-set' algorithm for the simulations of the energy landscape (Fig. 2, Supplementary Fig. 2 and Supplementary Fig. 3) and the compression test simulation (Fig. 3 and Supplementary Fig. 1) since a maximum step size can be defined that makes the tracking of all the hinge angles more reliable. The other simulations are performed using the 'SQP' (Sequential Quadratic Programming) algorithm. In this algorithm the step size is not limited and therefore converges faster to the energy minimum. Note that for this algorithm we fold the structures in 3 optimization steps.

**Compression test**. To verify the numerical model we performed an experimental compression test on a prismatic structure based on a cuboctahedron. Here, we describe the simulation of the compression test that we used as a comparison. In our simulations, we use a stiffness ratio of $\kappa = 10^{-4}$. We select two opposite faces on the structure, and apply constraints to the vertices that mimic the clamping in experiments. To do so, we only allow deformation of the vertices along the compression axis. We then create two additional edges ($S_{\text{comp}}$) parallel to the loading axis, and use them to connect both clamped faces. We use these edges to compress the structure by stepwise reducing their length and penalizing the energy according to

$$E_{\text{load}} = \sum_{i=1}^{S_{\text{comp}}} \frac{1}{2} k_{\text{p}} (l_i - L_i)^2, \tag{28}$$

where $k_{\text{p}}$ is the stiffness of the edges used to compressed the structure. We assign a stiffness that is much larger than the edge stretching stiffness and the hinge stiffness ($k_{\text{p}} \gg k_{\text{s}}, k_{\text{h}}$), so that effectively we are performing a displacement controlled compression.

In the simulation, we compress the structure in 1000 steps ($n_{\text{comp}}$), after which we stepwise remove the loading in the same number of steps. For each increment we allow the structure to relax with the specified constraints, while using previously mentioned optimization tool with the 'Active-set' algorithm.

To obtain the reaction force, $F$, during loading, we take the derivative of the elastic energy with respect to the displacement along the loading axis (z-axis), and find

$$F = \frac{dE}{dz} = \sum_{i=1}^{n_{\text{comp}}-1} \frac{E_{i+1} - E_i}{z_{i+1} - z_i}, \tag{29}$$

where we iterate through all the steps of the simulation to obtain the numerical gradient. Furthermore, we normalize the force by the weighted average of the stiffness defined by

$$\tilde{k} = \frac{k_{\text{s}}(S_E + S_D) + k_{\text{h}} H}{S_E + S_D + H}, \tag{30}$$

where $S_E$, $S_D$ and $H$ are the total number of edges, diagonals and hinges, respectively.

**Hinge selection reduction**. The prismatic structures studied here are highly symmetric, resulting from the underlying uniform polyhedra used as templates. Therefore, we developed a method to exploit these symmetries in order to reduce the search space to find stable states. Note that to determine unique selections of actuated hinges we consider only the edges of the internal polyhedron. In this section we will start from the polyhedron, and create a directed graph that represents the edges and their connectivity. Next, we determine the minimum distance matrix from the graph, from which we extract all principal sub-matrices that represent the hinge selections. In the remaining part of this section we explain our method in more detail, and as an example, apply it to a prismatic structure based on a triangular prism.

First, to determine symmetric hinge selections for the actuation of the prismatic structure, we start by constructing a graph of the internal polyhedron. Supplementary Fig. 9a shows an example of the graph belonging to a triangular prism, in which we have mapped the edges of the polyhedron to nodes in the graph. We then designate a type to each node in the graph, depending on the faces of the polyhedron that are connected to the corresponding edge, e.g., for a triangular prism there are two types: (a) triangle-square (denoted in red) and (b) square-square (denoted in purple).

We create directed connections between nodes in the graph. For this, we consider all the faces of the polyhedron, and define an outward-pointing normal

such that we can follow edges of each face using the right-hand-rule. For each pair of consecutive edges a connection is created between nodes in the graph, where the direction points from the first to the second node. For example, in Supplementary Fig. 9a, the edges numbered 8, 1, 7, and 4 form a face and are consecutive according to the described method, therefore directional connections from 8 → 1, 1 → 7, 7 → 4 and 4 → 8 are created.

Second, once the graph is created, we compute the shortest directed distance between the nodes. We use this distance to compare all the node selections, regardless of their original location on the prismatic structure. Symmetric node selections are characterized by similar paths traveled on the graph. To compute the shortest distance matrix we first made a depth search algorithm that enumerates all the possible paths between two nodes. Note that instead of using the length of the path in the matrix, which is typically done in graph theory, we use an array of the node types passed along that path (Supplementary Fig. 9c). In order to differentiate between equal length paths, we assign a numerical value to each one depending on the types of nodes and select the smallest number for consistency.

For example, for a triangular prism the function enumerates the paths between node 1 and 3. A few of the possible paths are $1 \rightarrow 7 \rightarrow 3$, $1 \rightarrow 2 \rightarrow 3$ and $1 \rightarrow 2 \rightarrow 8 \rightarrow 5 \rightarrow 4 \rightarrow 6 \rightarrow 7 \rightarrow 3$, indicated by the node type array *(aba)*, *(aaa)* and *(aabaaaba)*, respectively. We then assign the value 1 to the node type *a* and 2 to node type *b*. Two of the paths have the same length, but the assigned number representing the paths is different. We select path *(aaa)* that has the lowest value, 111. Therefore, the path on the shortest distance matrix, at entry $(1, 3)$ in Supplementary Fig. 9c, is *(aaa)* representing the path from node 1 through node 2 to node 3.

Last, from the shortest distance matrix we can compute the unique hinge selections after assigning values to the paths, by considering the eigenvalues of all the principal sub-matrices. We start by considering actuating one hinge, and after finding all unique hinge selections, increase the number of actuated hinges by one until the total number of hinges has been reached.

Since the distance between a node and itself is given by just the node type, we select one of each type. In case of a triangular prism we select nodes 1 and 7 with types *(a)* and *(b)*, respectively (Supplementary Fig. 9c). Next, we choose a second node expanding the previous selection of one node. We go through all the remaining nodes extracting the principal sub-matrices. These sub-matrices contain the information of the node types and the distance between nodes, and therefore their eigenvalues are unique to the node selection. Considering all the sub-matrices, we choose hinge selections that have different eigenvalues and remove duplication.

We repeat this strategy until all unique hinge selections have been found (Supplementary Fig. 9c). Note that we only need to consider hinge selections that contain up to half the number of nodes, given that we can simply consider the hinge selections to be reversed (i.e., replacing actuated hinges with non-actuated hinges and vice versa). For example, to choose eight nodes for a triangular prism, we inverse the one-node thus selecting nodes 2–9 and 1–6,8,9 which are the inverse of node 1 and 7, respectively.

**Stable states clustering**. In order to obtain the stable states that are unique under rotational or mirror symmetry, we consider the angles of all the hinges in the prismatic structure $n_{tot}$ in the final stable configuration, and place them in a $n_{tot}$-dimensional array, $\theta_{stable}$. We then separate the internal and external angles into two different groups. For each group we arrange the angles in order of increasing value. By doing so, we remove the spacial position of the hinges and obtain only the values of the angles in a specific order, thus considering all rotational and mirror symmetries. Next, we compare the ordered $\theta_{stable}$ arrays of all the stable state to cluster the ones that are similar to each other. For this we use a hierarchical clustering with centroid linkage[40] and a euclidean distance metric to shape the clusters. The maximum distance inside a cluster is of 1.5 rad. Therefore, we consider configurations with an average error of $\frac{1}{\sqrt{n_{tot}}}$ rad in every angle to be the same configuration. This error is up to 13.5 deg for the smallest structure based on a tetrahedron, and 7.8 deg for the structure based on truncated tetrahedron.

## Data availability
The data that support the findings of this study are available from the corresponding author upon reasonable request.

## Code availability
All computer algorithms necessary to reproduce the figures are available from the corresponding author upon reasonable request.

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

## Acknowledgements
We thank Alberico Sabbadini and Katia Bertoldi for initial discussions. This work is part of the Dutch Research Council (NWO) and was performed at the research institute AMOLF. It is part of the research program Innovational Research Incentives Scheme Veni from NWO with project number 15868 (NWO).

## Author contributions
A.I. and J.T.B.O. designed research; A.I., Y.L., and J.T.B.O. performed research; A.I. and J.T.B.O. analyzed data; and A.I. and J.T.B.O. wrote the paper.

## Competing interests
The authors declare no conflict of interest.
