## [Peer Review File · Nature Communications]

Reviewers' comments:

Reviewer #1 (Remarks to the Author):

The paper is on a type of reconfigurable prismatic architected materials consisting of extruded tubes mounted on the faces of polyhedrons. The concept was first appeared in Ref [9] of the paper (J. T. Overvelde, T. A. De Jong, Y. Shevchenko, S. A. Becerra, G. M. Whitesides, J. C. Weaver, C. Hoberman, and K. Bertoldi, A three-dimensional actuated 461 origami-inspired transformable metamaterial with multiple degrees of freedom, Nature Communications 7, 10929, 2016), and subsequently extended in Ref. [6] (J. T. Overvelde, J. C. Weaver, C. Hoberman, and K. Bertoldi, Rational design of reconfigurable prismatic architected materials, Nature 541, 347, 2017), both of which were written by the same authors. Hence, this would be the third paper on this type of structures. Obviously the paper has offered some new and detailed insight on the behavior of such structures. However, the concept itself is not sufficiently novel over the existing papers to warrant its publication as the third paper in Nature journals.

For the paper itself, I have the following comments to make.

(1) The idea shown in Fig. 1 appeared in Fig. 1(b) of Ref [6] and Fig. 1 of Ref [9]. What's new here?

(2) Ref. [9] gives a detailed account of the degrees of freedom for a unit with a cubic center. This is not the case in the current paper. When the center is replaced by a truncated cube, I suspect that the number of degrees of freedom will increase dramatically. Could the authors calculate the number? This will help to determine the number of actuators required to properly control the shape. This will also be important to determine whether some motions are rigid origami motions whereas others require deformation of facets. In Section "Reducing the search space", the authors simply consider all hinges to start with. This is due to the lack of analysis on the degrees of freedom.

(3) The authors assume that the faces of the structure are deformable by introducing edge and diagonal springs in order to identify minimum energy configurations. The detail of this approach is given in SI. The outline of the approach does not include any diagrams, making it hard to follow. For example, I cannot figure out Equation (4) in the SI which gives the angle between two faces connected by a rotational spring hinge. If n_a and n_b are the normal unit vectors of the two faces connected by the hinge, will θ be $\cos^{-1}(n_a \cdot n_b)$?

(4) The authors keep using the word "soft mode", e.g., lines 78 and 175. What is it?

(5) Assemblies consisting of many units are shown in Fig. 5. Again here all units are assumed to alter their shapes symmetrically. There are two issues here. First, the total number of degrees of freedom of the structure could be the sum of those of each unit in the worst case scenario (refer to my comment 2). Authors should discuss that. Second, with multiple degree of freedom system, non-symmetrical modes will be possible. The authors show none of them. Will they be impossible because of faces collide with each other?

(6) The authors show a single unit experiment with result matching that of prediction. The loading columns are rigidly fixed to the vertical tubes. However, as a material, it would be nice to see experiment on multi-unit structure (Ref. [9] shown that). My guess is that the result will be all over the places.

(7) In Fig. 1, it says the square faces are of 2.4mm. By watching the movie, it looks more like 2.4cm.

In conclusion, I think that this is a detailed paper about a concept that has been discussed twice

previously by the same group of authors. Though the concept is interesting, when it is extended to include hexagonal and octagonal shapes and above, the resultant structures will exhibit many degrees of freedom. Though there are stable positions, the whole structure will become quite “wobbly”. It will simply jump from one stable position to another under external loads while losing any load bearing capacity in-between. Without seeing the detailed discussion on this and physical models, I feel that the concept may be only valid in practice for the one shown in Ref. [9].

Reviewer #2 (Remarks to the Author):

I have found this paper very interesting and well-written and with a sufficient degree of novelty, mainly thanks to the rigorous numerical-analytical approach and the idea of the local actuation. The experiments are simple but conceptual. Thus I suggest to publish the paper as it is. (In fig 2 please comment on the absence in (a) of the small hysteresis present in (b))

Reviewer #3 (Remarks to the Author):

The manuscript “Exploring multistability in prismatic metamaterials through local actuation” explores how metamaterials can reconfigure. Such reconfigurability is crucial for realising materials that can be actuated in order to change their structure and properties. However, no general framework for understanding multistability exists because material-dependence and nonlinearities make the problem challenging. The manuscript presents a combinatoric path towards such a framework, complete with experimental verification of the major findings.

In general, the results presented are exciting and would be of broad interest to scientists and engineers interested in metamaterials, instabilities, and elasticity. Before I recommend publication in Nature Communications, I suggest the authors address the following remarks:

- I find that the manuscript could cite and discuss in more detail a broader range of relevant literature. The following works come to mind: “Recoverable and Programmable Collapse from Folding Pressurized Origami Cellular Solids” Li, Fang, Wang. Phys. Rev. Lett. 117, 114301 (2016) and “Self-folding origami at any energy scale” Pinson, Stern, Carruthers Ferrero, Witten, Chen, Murugan. Nature Communications 8, 15477 (2017).
- I found Figure 4 parts a and c difficult to follow. Do these present only numerical data (whereas b and d have experiments)? Also, the symbols and colour labels are not explained in the caption. I suggest expanding the caption for these parts.
- Could the authors comment on future work necessary for realising the exciting applications listed in the last sentence?
- There is a typo on line 161 and formatting issues with Ref. [8]

Response to the Referees' comments for manuscript
(Tracking #: NCOMMS-19-12693):
“Exploring Multistability in Prismatic Metamaterials Through
Local Actuation”

In the following, we address each of the Referees' comments/suggestions (*in italic*). In addition, while using the original numbering of figures/equations/references in the manuscript (*e.g.*, Fig. 1, Fig. 2,..., Eq. (1), Eq. (2),..., [1], [2], ...), we add prefix “R” for figures/equations/references presented in this response to the Referees' comments (*e.g.*, Fig. R1, Fig. R2, ... Eq. (R1), Eq. (R2), ..., Ref. [R1], Ref. [R2],...).

Response to Referee #1

Comment 1A

The paper is on a type of reconfigurable prismatic architected materials consisting of extruded tubes mounted on the faces of polyhedrons. The concept was first appeared in Ref [9] of the paper (J. T. Overvelde, T. A. De Jong, Y. Shevchenko, S. A. Becerra, G. M. Whitesides, J. C. Weaver, C. Hoberman, and K. Bertoldi, A three-dimensional actuated 461 origami-inspired transformable metamaterial with multiple degrees of freedom, Nature Communications 7, 10929, 2016), and subsequently extended in Ref. [6] (J. T. Overvelde, J. C. Weaver, C. Hoberman, and K. Bertoldi, Rational design of reconfigurable prismatic architected materials, Nature 541, 347, 2017), both of which were written by the same authors. Hence, this would be the third paper on this type of structures. Obviously the paper has offered some new and detailed insight on the behavior of such structures. However, the concept itself is not sufficiently novel over the existing papers to warrant its publication as the third paper in Nature journals.

We thank the Referee for his/her comments. It is true that we previously studied the same prismatic building blocks for which we have already introduced a general design approach [6] and a pneumatic actuation strategy [9]. However, in the current study we reconsider the behavior of these structures and for the first time are able to observe highly non-linear effects such as multistability. In contrast to [6] in which we assumed a highly simplified model based on rigid origami and considered only small deformations (mode analysis), we now explore the effect of additional deformation of the faces (not studied before) and analyze their full geometrical nonlinear behavior (large changes in angle). Our numerical model and the local actuation approach that we introduce in this paper allow us to shed light on the complex energy landscape of these structures and their ability to maintain multiple stable states. Moreover, a similar approach could be used to study the non-linear behavior of other types of metamaterials.

As such, the current article introduces a method to study the more complex and interesting multistable behavior of these prismatic structures that has not yet been considered, which is essential to bring these structures closer to practical conditions and implementation. Based on this and the comments below we have updated the introduction of the paper and highlighted the difference with the previous published papers:

“As such, here we study a class of prismatic multistable 3D building blocks, that are based on polyhedra templates. These building blocks have previously been studied from an infinitesimal deformation and rigid origami framework [6], however here we assume that these structures can undergo large rotations and deformations of the faces, making the energy of these structures highly non-linear and thus significantly more difficult to explore. To do so, we introduce a numerical method to search for energy minima that correspond to the stable states of the prismatic structures. While a complete description of all possible deformations and stable states is not possible due to the large number of degrees of freedom arising from the elastic description, our method was designed to closely mimic possible experimental implementations of locally actuated metamaterials previously studied for only one prismatic structure [9]. As a result, we are able to shine light on the highly multistable behavior that most of these building blocks exhibit. We start by introducing the design of the 3D building blocks and a numerical model to simulate their behavior. We next validate our numerical approach with centimeter-scale prototypes. In order to gain insight in the problem, we then visualize the non-linear energy landscape of multiple prismatic building blocks by applying local actuation to two hinges. Based on these results, we develop a method to extract all possible unique actuation patterns, allowing us to efficiently scan through the energy landscape and find additional stable configurations. Finally, we show for a few multistable building blocks that they can be tessellated to create multistable metamaterials.”

Comment 1B

For the paper itself, I have the following comments to make.

(1) The idea shown in Fig. 1 appeared in Fig. 1(b) of Ref [6] and Fig. 1 of Ref [9]. What’s new here?

As discussed in previous comment 1A, the concept of multistability in these prismatic structures has not yet been considered before. To highlight this, we have updated Fig. 1 (Fig. R1) and included the contrast with previous studies. We have also updated the caption to make this point more clear.

Comment 1C

(2) Ref. [9] gives a detailed account of the degrees of freedom for a unit with a cubic center. This is not the case in the current paper. When the center is replaced by a truncated cube, I suspect that the number of degrees of freedom will increase dramatically. Could the authors calculate the number? This will help to determine the number of actuators required to properly control the shape. This will also be important to determine whether some motions are rigid origami motions whereas others require deformation of facets. In Section “Reducing the search space”, the authors simple consider all hinges to start with. This is due to the lack of analysis on the degrees of freedom.

Under the same framework of [6,9] (assuming rigid origami where the facets cannot stretch, shear or bend) as the Referee mentions, a prismatic structure with a cubic center has 3 degrees of freedom.

Figure R1: Prismatic structures and their multiple stable states. The prismatic structures can be designed by extruding the edges of a convex polyhedron perpendicular to the faces. The multistable examples shown here are based on a) a truncated tetrahedron, b) a truncated cube and c) a cuboctahedron. The additional stable states can only be reached by going over a finite energy barrier resulting from deformation of the faces of the structure. The prototypes have square faces of 24 mm made from cardboard (0.4 mm) and connected through hinges made from double-sided tape [6].

However, as reported in [6], under the same assumptions the truncated cube is rigid, and does not show any degrees of freedom. By changing the framework to elastic origami, we add many degrees of freedom (nodal displacements) to the structures, since elastic origami assumes that not only the hinges can bend but the facets can deform. This represents a more realistic model, allowing us to better understand to what extent the assumptions of rigid origami are valid. However, this makes an analytical description of all the deformations nearly impossible, as also mentioned in the paper. While under the assumption of elastic origami the structure still has many degrees of freedom, some deformations require significantly less energy than others. We refer to these deformations as soft modes (because they still require energy to deform) instead of degrees of freedom. In general these soft modes are related to the degrees of freedom found under the assumption of rigid origami.

To show the effect of face deformation on the structural response, we have updated the article and now mention the degrees of freedom of the prismatic structures that arise from the constraints upon assuming rigid origami throughout the main text and in the SI.

Additionally by assuming that the hinges have non-zero stiffness, for all the structures the initial extruded state corresponds to an energy minimum and is therefore stable. Thus, the shapes presented in this work do not correspond to soft modes, but always to energy minima. To deform these structures away from their initial position, we need actuation regardless if it is along a soft mode or if it requires significant deformation of the facets. Therefore, in Section “Reducing the search space” we consider all possible hinge combinations, not just those of the soft modes.

We updated this section in the paper to clarify our approach:

“We start by deriving the number of hinge combinations that can be actuated. For these combinations we consider all hinges of the structure, because by definition deformation of the structure requires energy (e.g. actuation) regardless whether the structure exhibits soft modes or not. Each prismatic structure is composed of two hinge types: internal and external. The internal hinges correspond to the edges of the polyhedron that is used

as a template, while the external hinges arise from the extrusion process.”

Comment 1D

(3) *The authors assume that the faces of the structure are deformable by introducing edge and diagonal springs in order to identify minimum energy configurations. The detail of this approach is given in SI. The outline of the approach does not include any diagrams, making it hard to follow. For example, I cannot figure out Equation (4) in the SI which gives the angle between two faces connected by a rotational spring hinge. If n_a and n_b are the normal unit vectors of the two faces connected by the hinge, will theta be $\cos^{-1}(n_a \cdot n_b)$?*

We agree with the Referee, and have included a diagram in the SI to indicate location of the springs and the mentioned vectors from Eq. (4) in the SI (Fig. R2). While the referee is correct in pointing out that the angle between two faces can be defined according to $\theta = \cos^{-1}(n_a \cdot n_b)$, here we used a slightly different representation since the domain of the function \cos^{-1} is limited between -1 and 1. Therefore, we use the Four-quadrant function \tan^{-1} for which the domain is given from $-\infty$ to ∞ . With this inverse tangent function we have less convergence problems compared to the inverse cosine function. Additionally it defines the angle over the full range of $-\pi$ to π . Besides adding the schematic, we have updated the SI to indicate this:

“Each hinge angle can be found from the coordinates of the vertices according to

$$\theta = \tan^{-1} \left(\frac{\mathbf{a}_h \cdot (\mathbf{n}_a \times \mathbf{n}_b)}{\mathbf{n}_a \cdot \mathbf{n}_b} \right), \tag{R1}$$

in which \mathbf{n}_a and \mathbf{n}_b are the normal vectors of the two faces connected by the hinge, and the vector \mathbf{a}_h lies along the hinge axis (Fig. R2a). Note that we use the function \tan^{-1} (instead of e.g. using $\theta = \cos^{-1}(n_a \cdot n_b)$) since its domain is defined for $(-\infty, \infty)$ and the angle can be defined between $[-\pi, \pi]$ using a Four-quadrant inverse tangent.”

Figure R2: Schematic of the extruded unit cell [6].

Comment 1E

(4) *The authors keep using the word “soft mode”, e.g., lines 78 and 175. What is it?*

We are fully aware that the terminology ‘soft mode’ is not used in all communities (See also the answer to Comment 1C). Therefore, we modified the text to explain this definition better:

“Each polyhedron in the tessellation is used as a basis for a thin-walled building block, that is constructed by extruding the edges of the polyhedron in the direction normal to the corresponding face (Fig. R1). When assuming rigid origami [25] (i.e. the structure can only fold along predefined hinges), some of the building blocks cannot change shape (Fig. R1a-b), while others can be reconfigured along specific degrees of freedom (Fig. R1c). Interestingly, for all of these examples we found additional stable configurations that are spatially admissible, but that cannot be reached without temporarily deforming the rigid faces (Fig. R1). These states correspond to minima in elastic energy that are separated by infinitely high barriers, i.e. they are topologically isolated [26]. By allowing the faces to stretch or bend (i.e. elastic origami) [14], we lower the energy barrier such that moving between local minima becomes kinetically admissible. Note that under the assumption of elastic origami the structure has many degrees of freedom, however, some deformations require significantly less energy than others, corresponding to the degrees of freedom obtained when assuming rigid origami. We refer to these deformations as soft modes instead of degrees of freedom. While for some simple origami patterns the energy of the system can be computed analytically [12,13,27], already a generic degree-four vertex pattern becomes nearly impossible to decipher [16].”

Comment 1F

(5) Assemblies consisting of many units are shown in Fig. 5. Again here all units are assumed to alter their shapes symmetrically. There are two issues here. First, the total number of degrees of freedom of the structure could be the sum of those of each unit in the worst case scenario (refer to my comment 2). Authors should discuss that. Second, with multiple degree of freedom system, non-symmetrical modes will be possible. The authors show none of them. Will they impossible because of faces collide with each other?

The Referee is correct in pointing out that the response of the material (i.e under the assumption of periodic boundary conditions) could potentially be the same as the unit cell. However, we find that the periodic boundary conditions introduce additional constraints that considerably limits the stable states that can be achieved (as was also reported in [6] under the assumption of rigid origami) and has a huge effect on the non-linear response of the material. In fact, many of the asymmetric modes that we found for the unit cells can no longer be achieved in the materials. Therefore, the challenge here is to design materials that are capable of being mechanically stable in additional stable states. Here we show an initial step to find these materials in the current non-linear framework. A more detailed study that involves periodic boundary conditions with larger periodicity could help this purpose, but falls outside the scope of the current article.

We have highlighted these considerations in the article when discussing the results of the multi-stable materials:

“We have shown that by local actuation we are able to effectively explore the non-linear energy landscape of prismatic structures and find additional minima. We next show that our method can also be applied to find stable states in prismatic metamaterials assembled from these building blocks, with periodic boundary conditions (SI: *Minimization of energy*). Note that the periodic boundary conditions introduce additional constraints that considerably limits the stable states that can be achieved and has a huge effect on the non-linear response of the material. In fact, many of the asymmetric stable states that we found for the unit cells can no longer be achieved in the materials. Therefore, the challenge here is to design materials that are capable of being mechanically stable in

additional states. To perform a first exploration, we focus on cubic tessellations that can be constructed from a single prismatic structure. We only consider the 11 polyhedra that were previously found to be multistable, which results in a total of 15 materials (Fig. S7). In order to search for stable states and to limit the number of actuation patterns that have to be applied, we apply the same actuation patterns to the material that previously resulted in stable configuration for the prismatic structure used as a unit cell.”

Comment 1G

(6) The authors show a single unit experiment with result matching that of prediction. The loading columns are rigidly fixed to the vertical tubes. However, as a material, it would be nice to see experiment on multi-unit structure (Ref. [9] shown that). My guess is that the result will be all over the paces.

As shown in Ref [6,9], a material constructed from many unit cells keeps its shape despite having several degrees of freedom. To deform the structure along their soft modes a force needs to be applied. Importantly, many of the metamaterials found here do not exhibit soft modes and have to overcome energy barriers to move (which can be tuned by changing the ratio κ). Therefore, these structures are inherently more stable than the structures explored previously in Ref [6,9]. We believe that previous results and the results shown here on the building blocks provide enough insight into the potential behavior of the metamaterials. In fact, as also discussed in Comment 3D, the fabrication of such materials with the right experimental parameters at smaller scales and with many unit cells is still a challenge that requires future work. To discuss this, we have updated the conclusion of the paper to point out several techniques that can solve such problem such as 2 photon lithography or stereolithography to fabricate these materials. Moreover we have included additional experiments following comment 1I.

Comment 1H

(7) In Fig. 1, it says the square faces are of 2.4mm. By watching the movie, it looks more like 2.4cm.

It was not our intention to mislead the Referees as also indicated by the movies in which we show the relative size of the structures compared to our hands. We have updated the paper and fixed this mistake. Additionally we have clarified the dimensions of our prototypes throughout the paper.

Comment 1I

In conclusion, I think that this is a detailed paper about a concept that has been discussed twice previously by the same group of authors. Though the concept is interesting, when it is extended to include hexagonal and octagonal shapes and above, the resultant structures will exhibit many degrees of freedom. Though there are stable positions, the whole structure will become quite “wobbly”. It will simply jump from one stable position to another under external loads while losing any load bearing capacity in-between. Without seeing the detailed discussion on this and physical models, I feel that the concept may be only valid in practice for the one shown in Ref. [9].

We hope that our answers to the Referee’s comments has convinced him/her that our paper focuses on new behavior not discussed previously. As indicated by the Referee, some of the structures show

soft modes (indicated by the Referee as “wobbly”), which was the focus of the previously published articles [6,9]. Here, we show that we can still deform the structures through face deformations that require more work, even if the structure does not show any soft modes. Interestingly, the energy barrier that needs to be overcome to jump between stable states can be tuned by changing the ratio between hinge and face stiffness κ , as also discussed in the paper (Fig. 4). Therefore, stability of the stable states of these structures can be tuned and adjusted so external loads do not affect this stability.

However, the Referee is also correct in pointing out that only a few smaller structures have been tested (Fig. 4b and d and Movie S3 show experiments on prismatic structures based on a truncated tetrahedron and cube). Therefore, we have included additional experiments on a larger prismatic structure based on a truncated cube and a rhombicuboctahedron in the supplemental information (Fig. R3)

Finally, we thank the Referee again for these comments, and feel that by updating the paper according to his/her comments we have been able to clarify the scope of current work compared to previous work.

Figure R3: Comparison of selected stable states found with our numerical algorithm (Fig. S6) and experiments for two prismatic structures based on a) a truncated cube and b) a rhombicuboctahedron.

Response to Referee #2

Comment 2A

I have found this paper very interesting and well-written and with a sufficient degree of novelty, mainly thanks to the rigorous numerical-analytical approach and the idea of the local actuation. The experiments are simple but conceptual. Thus I suggest to publish the paper as it is.

We thank the Referee for his/her positive feedback.

Comment 2B

In fig 2 please comment on the absence in (a) of the small hysteresis present in (b).

The Referee is correct in pointing out that at the end of the compression in experiments for a prismatic structure based on a cuboctahedron, we do not observe a small hysteresis loop that we do get in simulations. This hysteresis is the result of a sharp folding of some of the sides of the structure, which do appear in experiments. However, the lack of hysteresis can be a result of small misalignment in experiments, small fabrication errors of the prototype, etc. Importantly, the model does qualitatively predict most of the behavior observed in experiments. We have updated the caption of the paper to mention this difference:

“The behavior of both the experiments and the simulations are strikingly similar, both showing one instability when compressing the structure and two when returning to the initial configuration. Note that a small hysteresis loop observed in simulations at the end of the compression is not captured in experiments possible due to a small misalignment when clamping the structure or small fabrication errors of the prototype.”

Response to Referee #3

Comment 3A

The manuscript “Exploring multistability in prismatic metamaterials through local actuation” explores how metamaterials can reconfigure. Such reconfigurability is crucial for realising materials that can be actuated in order to change their structure and properties. However, no general framework for understanding multistability exists because material-dependence and nonlinearities make the problem challenging. The manuscript presents a combinatoric path towards such a framework, complete with experimental verification of the major findings.

In general, the results presented are exciting and would be of broad interest to scientists and engineers interested in metamaterials, instabilities, and elasticity.

We thank the Referee for his/her positive feedback and comments, and have addressed all the remarks below.

Comment 3B

Before I recommend publication in Nature Communications, I suggest the authors address the following remarks: I find that the manuscript could cite and discuss in more detail a broader range of relevant literature. The following works come to mind: “Recoverable and Programmable Collapse from Folding Pressurized Origami Cellular Solids” Li, Fang, Wang. Phys. Rev. Lett. 117, 114301 (2016) and “Self-folding origami at any energy scale” Pimson, Stern, Carruthers Ferrero, Witten, Chen, Murugan. Nature Communications 8, 15477 (2017).

Following the suggestion of the Referee, we have included additional literature to our article. We have updated the introduction according to:

“A challenging problem has been to design multifunctional materials, i.e. materials that can vary their properties. So far, this has been achieved by e.g. taking inspiration from origami to create internal structures that can be reconfigured along a few degrees of freedom [6-8]. Finding the structure of such reconfigurable materials is not trivial since the number of degrees of freedom for a general origami design grows exponentially [R1], and typically a general design approach [R2] is needed to satisfy required conditions [R3]. Once created, these materials exhibit highly anisotropic behavior, enabling the change of their properties by applying locally a range of stimuli including air pressure [9,10], pre-stresses [8] and swelling [11]. However, the deformed state of these materials becomes dependent on these stimuli, and once they are removed the material will relax to the initial configuration.”

and

“A way to overcome this dependency is to introduce multistability [12-17]. This can be achieved by having two or more stable states that differ in configuration and are separated by significant energy barriers. Multistability has already been used to create auxetic [2,8] and energy trapping metamaterials [19,20,R4] as well as deployable [17,21], morphing [22] or crawling [23] structures, however, most of these materials are assembled from 2D building blocks that can switch between only two stable states. A natural question to

ask is whether 3D building blocks with more than two stable configurations exist and if they can be used to form multifunctional metamaterials.”

Comment 3C

I found Figure 4 parts a and c difficult to follow. Do these present only numerical data (whereas b and d have experiments)? Also, the symbols and colour labels are not explained in the caption. I suggest expanding the caption for these parts.

To clarify the results shown in this figure, as suggested we have expanded the caption of the figure:

“Comparison of stable states found with our numerical algorithm and experimental verification of a selection of the results for two prismatic structures. The normalized stretch energy ($\tilde{E}_{\text{stretch}}$) and maximum stiffness ratio (κ_{max}) were obtained with simulations for the unique stable states of a prismatic structure based on a) a truncated tetrahedron and c) a cube. Blue dots indicate stable states found by actuation of up to 3 hinges simultaneously, while the yellow dots indicate all other stable states found by actuating more than 3 hinges. The pink and purple colors indicate regions with stable states that were achieved in experiments. b) The stable states of the structures based on a truncated tetrahedron are compared to experiments using a prototype with 3D printed faces and Mylar hinges. d) The stable states of the structures based on a cube cannot be obtained with prototypes made with Mylar hinges. Instead, more stretchable silicon hinges are used to obtain the stable states. The arrows indicate that the state is not stable in experiments, but relaxes to a different stable state. The single asterisk (*) indicates that the stable state could not be achieved in experiments due to a limited maximum stretch, while the double asterisk (**) indicates that the state cannot be achieved due to non-adjacent face crossing.”

Comment 3D

Could the authors comment on future work necessary for realising the exciting applications listed in the last sentence?

While we believe that our work can be used in interesting applications such as those mentioned in the paper (reconfigurable acoustic wave guides, microelectronic mechanical systems and energy storage devices), this current work has been mostly exploratory and focused on a numerical method to study the multistable behavior, demonstrated by proof-of-concept experiments. Future work is needed to bring these mechanical concepts and designs closer to the mentioned (and other) application. The main challenges lie at the fabrication of these 3D samples at the micro-nano scale. There has already been significant progress in the fabrication of origami-like materials at these scales, and we believe that similar methods can be used to build our proposed structures. An important aspect will also include the development of suitable actuation strategies. Note that if we would like to address all stable states individually, local actuation would be required. However, global stimuli could still potentially lead to different folding behavior for a single structure by incorporating materials that respond differently to stimuli, or by varying the loading rate. We have included this additional discussion in our conclusion:

“While we have only validated our experiments at the centimeter-scale using relatively simple fabrication techniques, the mechanical behavior of our systems is theoretically

scale-independent. However fabricating such intricate structures at the micro/nano scale is not trivial. More advanced 3D-fabrication techniques such as two-photon lithography [31] or stereolithography [32] could lead this innovation. Additionally, scaling off these structures will introduce additional environmental influences such as capillary effects. It is not known how such forces influence the (multistable) behavior of our structures, and additional research is required to explore this direction. Moreover, instead of manually deforming the structure, they can be made responsive by applying local actuation to the hinges. For our prototypes, this can be achieved by applying pneumatic pouches to some of the hinges [9]. As an example, already by discrete actuation of two pouches applied to the prismatic structure based on a truncated tetrahedron, we were able to achieve four of the stable states (Movie S5). Similar strategies could be applied to actuate our designs at smaller length scales, using e.g. localized swelling of hydrogels for actuation. However, individually addressable local actuation patterns become difficult to apply at small scales and therefore a particular interesting future direction is the application of global stimuli to trigger the multistable behavior, e.g. heat or pH variations. Global stimuli can potentially lead to different folding behavior for a single structure by incorporating multiple materials into the design that each respond differently to changes in their environment or to different loading rates. Therefore, we believe that these prismatic multistable materials could lead to the next generation of multifunctional metamaterials that can be applied as e.g. reconfigurable acoustic wave guides [33], microelectronic mechanical systems [34] and energy storage devices [35].”

Comment 3E

There is a typo on line 161 and formatting issues with Ref. [8]

We thank the referee for carefully looking through the text. We have updated the text according to his/her comments and corrected additional typos that we found on the text.

References

- [R1] B. G.-g. Chen and C. D. Santangelo, Branches of triangulated origami near the unfolded state, *Phys. Rev. X* **8**, 011034 (2018).
- [R2] M. B. Pinson, M. Stern, A. Carruthers Ferrero, T. A. Witten, E. Chen, and A. Murugan, Self-folding origami at any energy scale, *Nature Communications* **8**, 15477 (2017).
- [R3] T. Tachi and T. C. Hull, Self-Foldability of Rigid Origami, in *Volume 5B: 40th Mechanisms and Robotics Conference*, Vol. 9 (American Society of Mechanical Engineers, 2016) p. V05BT07A029.
- [R4] S. Li, H. Fang, and K. W. Wang, Recoverable and programmable collapse from folding pressurized origami cellular solids, *Phys. Rev. Lett.* **117**, 114301 (2016).

Reviewers' comments:

Reviewer #1 (Remarks to the Author):

The authors have addressed some of my concerns and the revised paper is much clearer than the first submission. I appreciated their hard work.

However, I am not fully satisfied with some of the answers provided by the authors. I would like to make a few further comments.

1. When a basic unit is rigid, as in the case of (a) and (b) in Fig. 1, will additional degrees of freedom appear after a few units are combined together even if facets are rigid but hinges are free to rotate? Similarly, if the basic unit is not rigid, as in © of Fig. 1, will additional degrees of freedom appear after a few units are combined together even if facets are rigid but hinges are free to rotate? How many total degrees of freedom will appear in rigid origami sense?

A discussion will be useful. Or the authors could confine themselves to cases where there is no degrees of freedom in rigid origami sense.

2. The work presented in this paper is mainly about the behavior of single units. Only computer simulations are shown which predict the behavior of assemblies of such units. The authors further mention that some of the asymmetrical soft modes will vanish as the units are assembled together. They also mention that it is difficult to make small scale models, and hence, no such things are made to experimentally validate their findings.

I entirely appreciate that it is not feasible in this paper to discuss how the structures can be made at very small scale, and I will not ask for that. However, the authors have made individual units. Is it possible to make a relatively large assembly with centimeter scale units (which they have already have) to validate their findings? They have shown that adjusting the ratio between the hinge stiffness and face stiffness can change the behavior of the entire assembly. Why not try it on an assembly?

3. The work is mainly about a family of structures that can alter their shapes should appropriate loads (actuation) are applied. In my previous comments, I mentioned "It (the assembly) will simply jump from one stable position to another under external loads while losing any load bearing capacity in-between. Without seeing the detailed discussion on this and physical models, I feel that the concept may be only valid in practice for the one shown in Ref. [9]." The authors seem not answering this concern. Perhaps the authors should be straightforward on this by saying that these structures are only good for shape changing materials/structures.

4. Some papers listed by the authors have discussed energy change when both hinges and the faces are elastic. These structures, often folded from a flat sheet with certain crease patterns, have been analyzed both analytically with simplified models and numerically. Hence, this analysis is not something new. Applying this to the structures reported here and those in [6] and [9] are something new. Again it will be good if this is clear.

Reviewer #3 (Remarks to the Author):

The authors have addressed my comments well. The physical phenomena explored in this work are fundamentally new, and can lead to novel types of materials with important applications. I therefore recommend the publication of the present version of this manuscript.

Minor comment: On Line 435, "off" should be changed to "of"

Response to the Referees' comments for manuscript
(Tracking #: NCOMMS-19-12693A):
“Exploring Multistability in Prismatic Metamaterials Through
Local Actuation”

In the following, we address each of the Referees' comments/suggestions (*in italic*). In addition, while using the original numbering of figures/equations/references in the previous version of the manuscript (*e.g.*, Fig. 1, Fig. 2,..., Eq. (1), Eq. (2),..., [1], [2], ...), we add prefix “*R*” for figures/equations/references presented in this response to the Referees' comments (*e.g.*, Fig. R1, Fig. R2, Eq. (R1), Eq. (R2), ..., Ref. [R1], Ref. [R2],...).

Response to Referee #1

Comment 1A

The authors have addressed some of my concerns and the revised paper is much clearer than the first submission. I appreciated their hard work. However, I am not fully satisfied with some of the answers provided by the authors. I would like to make a few further comments.

We thank the Referee for his/her comments, and hope that with this second round we have been able to appropriately answer the comments of the Referee.

Comment 1B

1. When a basic unit is rigid, as in the case of (a) and (b) in Fig. 1, will additional degrees of freedom appear after a few units are combined together even if facets are rigid but hinges are free to rotate? Similarly, if the basic unit is not rigid, as in (c) of Fig. 1, will additional degrees of freedom appear after a few units are combined together even if facets are rigid but hinges are free to rotate? How many total degrees of freedom will appear in rigid origami sense? A discussion will be useful. Or the authors could confine themselves to cases where there is no degrees of freedom in rigid origami sense.

When the structure is rigid (under the assumption of rigid origami), connecting multiple of these structures together will not introduce additional degrees of freedom. This is because the connections between structures are rigid. On the other hand, if a structure has degrees of freedom (when still assuming rigid origami), connecting multiple structures together would add up their degrees of freedom, but only when no periodic boundary conditions are assumed or until a certain connectivity is reached. For example if we connect two structures with a single degree of freedom, the resulting

structure will have two degrees of freedom. However, for the structure based on a cube the degrees of freedom do not scale with the amount of unit cells connected, and depends on the connectivity of the unit cells. An analysis of this case is given in the supplemental data of previous work [12]. Specifically, edge modes or corner modes can appear, which results from the reduced connectivity (i.e. number of constraints) at the boundary. For an example of this phenomenon see the supplemental data of previous work [6]. Importantly, the exact number of degrees of freedom that will appear depends greatly on the assembly. Previous efforts to introduce a counting argument were not successful and did not reveal any particular set of guidelines. Therefore, to really understand the final number of degrees of freedom that appears, each assembly needs to be considered individually.

Furthermore, when no longer considering rigid origami these effects will still appear. Moreover, additional degrees of freedom might appear with wavelengths longer than the size of the unit cell, especially in case when non-linearities are taken into account. Taking all these effects into consideration and to simplify the analysis, we restricted ourselves to study these assemblies with periodic boundary conditions applied to the smallest possible unit cell. Additionally, we do not want to constrain ourselves to cases where there are no degrees of freedom in rigid origami sense, given that we find multistability in all the unit cells we analyzed, independent of the amount of soft modes they have or the need of face deformation to change shape.

We have included a description of the assumptions that we made for our periodic material in the article, to clearly demonstrate the applicability and limits of our current analysis. See the response to comment 1C for the updated text.

Comment 1C

2. The work presented in this paper is mainly about the behavior of single units. Only computer simulations are shown which predict the behavior of assemblies of such units. The authors further mention that some of the asymmetrical soft modes will vanish as the units are assembled together. They also mention that it is difficult to make small scale models, and hence, no such things are made to experimentally validate their findings. I entirely appreciate that it is not feasible in this paper to discuss how the structures can be made at very small scale, and I will not ask for that. However, the authors have made individual units. Is it possible to make a relatively large assembly with centimeter scale units (which they have already have) to validate their findings? They have shown that adjusting the ratio between the hinge stiffness and face stiffness can change the behavior of the entire assembly. Why not try it on an assembly?

Following the suggestion of the Referee, we have fabricated an assembly of a prismatic structures based on a cuboctahedron, as shown in Fig. R1e-g. The assembly consists of 8 unit cells in a $2 \times 2 \times 2$ arrangement. If we first compare the simulation results of the stable states of the unit cell and the corresponding periodic tiling, we find that many of the stable states disappear (Fig. R1d). In fact, for the periodic assembly we find that more stretching energy is needed in the three feasible stable states, which indicates that the stable states are not equal to the ones of the prismatic building blocks.

Note that previous experiments on a single unit cell with hinges made from $50 \mu\text{m}$ thick Mylar sheets predicted a $\kappa \approx 10^{-3}$, such that all states with $\kappa_{\text{max}} > 10^{-3}$ are attainable. Similarly, we find for our prototype of the assembled metamaterial that, of the three potentially stable states, only one stable state has $\kappa_{\text{max}} > 10^{-3}$ (Fig. R1e). As an additional check, we fabricated an additional prototype for which we increased the thickness of the Mylar hinges to $125 \mu\text{m}$. As expected, this increases κ_{max} , such that no stable states exists anymore, as shown in the additional supplemental movie (Movie S5).

Relating back to Comment 1B, we can now also use the assembled metamaterial to show the limits of our analysis and the assumptions we made. In particular, we find that additional stable states can be found for the metamaterial that are either not periodic (i.e. they are related to edge modes as previously found [6]), or that exhibit a periodicity that requires a larger wavelength (i.e. the deformation is not the same for each unit cell, but could still be modeled using periodic boundary conditions when using a unit cell made from more than one prismatic building block) shown in Fig. R1f and g, respectively. While including a full analysis falls outside the scope of current paper, we did include these results and a discussion on the current limitations of our analysis in the paper. We have updated the main text of the paper according to:

“We find that most (11) of the materials show additional stable states (Fig. S9). In Fig. R1a-c and Movie S4 we show three of these multistable materials and some of their stable states with unit cells based on a cuboctahedron, a rhombicuboctahedron and a truncated cuboctahedron. Note that the energy of the metamaterial’s stable states do not have to be equal to the prismatic building blocks. This can be seen for example from the stretch energies of the unit cell of a cuboctahedron in Fig. R1d, which have shifted for their periodic arrangement.

To validate these findings, we fabricated a material assembly based on a cuboctahedron, that contains $2 \times 2 \times 2$ building blocks. As expected, we are only able to achieve one of the stable states (Fig. R1e), since states *ii* and *iii* lie below the $\kappa_{\max} \approx 10^{-3}$ threshold as previously predicted. To verify this, we also fabricated a second sample with thicker Mylar sheets of 125 μm , and as expected we found that this metamaterial does not exhibit any stable states (Movie S5).

While here we have limited our analysis to periodic boundary conditions applied to the unit cells, similar as in rigid origami [?], stable states can appear on the edges and corners of the materials. This is clearly demonstrated in Fig. R1f showing an edge state. We are also able to observe additional stable states with wavelength longer than the unit cells (Fig. R1g). In order to capture these in our model larger unit cells would have to be considered. Therefore, additional studies have to be performed to continue exploration of their rich energy landscape, including many other tessellations that can be constructed.”

And added the following movie description in the SI:

Movie S5. Varying the multistable behavior in a metamaterial. We control the multistability of a metamaterial based on a cuboctahedron by adjusting κ . In experiments, this can be achieved by changing the hinge thickness. When we build our prototypes with Mylar hinges with a thickness of 50 μm , the metamaterial exhibits multiple stable states. However, when building the material with 125 μm thick Mylar hinges it is no longer multistable.

Comment 1D

3. The work is mainly about a family of structures that can alter their shapes should appropriate loads (actuation) are applied. In my previous comments, I mentioned "It (the assembly) will simply jump from one stable position to another under external loads while losing any load bearing capacity in-between. Without seeing the detailed discussion on this and physical models, I feel that the concept may be only valid in practice for the one shown in Ref. [9]." The authors seem not answering this concern. Perhaps the authors should be straightforward on this by saying that these structures are

Figure R1: Simulations and experiments of multistable metamaterials. The simulated metamaterials are based on cubic tessellations of prismatic structures based on a) a cuboctahedron, b) a rhombicuboctahedron and c) a truncated cuboctahedron. d) Normalized stretch energy ($\tilde{E}_{\text{stretch}}$) and maximum stiffness ratio (κ_{max}) for the unit cell and metamaterial based on a cuboctahedron. e) One of the stable states of the prismatic structure based on a cuboctahedron found in simulations can also be achieved in a prototype of the metamaterial made with 50 μm thick Mylar hinges. Additionally, we find f) edge and corner states and g) stable states with periodicity larger than a single unit cell not covered by our simulations.

only good for shape changing materials/structures.

Our structures can indeed change shape from one stable state to another, however, the deformation between the states is not stable when removing the load. For the stable configurations, deformations require energy due to face stretching, which by definition means that they can bear load. We hope this has become clear from the additional experiments performed in Comment 1C.

Note that in general the magnitude of the load depends greatly on the energy barrier between stable states. Indirectly, this is described by the value of κ_{\max} for each stable state, as shown in Fig. 4 and Fig. S7. Higher values require more perturbations, such that we expect the structures to be more stable. Moreover, this energy barrier can be tuned by changing the stiffness ratio κ and therefore the load-bearing capacity at any particular stable deformation can be increased such that effectively no external load affects the material's stability. Following this, we believe that e.g. the statement on load bearing mentioned in the conclusions "Additionally, by tessellating these prismatic structures, we find multistable metamaterials that can reconfigure their architecture and therefore tune their properties. Importantly, these materials do not require energy to maintain their stable configurations, and will be robust to external variations as significant energy barriers have to be overcome to alter their structure." is valid.

Comment 1E

4. Some papers listed by the authors have discussed energy change when both hinges and the faces are elastic. These structures, often folded from a flat sheet with certain crease patterns, have been analyzed both analytically with simplified models and numerically. Hence, this analysis is not something new. Applying this to the structures reported here and those in [6] and [9] are something new. Again it will be good if this is clear.

We completely agree with the Referee that using springs to model both rigid origami as well as origami that can exhibit face stretching or bending is not new. However, to the best of our knowledge, our approach that uses local actuation to explore the energy landscape, in combination with the prismatic structures, has not yet been introduced. To clarify that the description of the elastic energy is not new, we have updated the manuscript:

"To model the thin-walled prismatic structure as shown in Fig. 1, we define the elastic energy of the prismatic structures using linear springs similar to previous work [16,17,32] (SI; Minimization of energy)..."

Response to Referee #3

Comment 3A

The authors have addressed my comments well. The physical phenomena explored in this work are fundamentally new, and can lead to novel types of materials with important applications. I therefore recommend the publication of the present version of this manuscript.

Minor comment: On Line 435, "off" should be changed to "of"

We thank the Referee for his/her positive comments, and have changed the minor error.

REVIEWERS' COMMENTS:

Reviewer #1 (Remarks to the Author):

The authors have addressed my concerns and I am happy for it to be published. I appreciate that a larger model is made and it shows the behaviour that the authors predicted.